# Learning Deep Generative Models of Graphs

## Abstract

Graphs are fundamental data structures required to model many important real-world data, from knowledge graphs, physical and social interactions to molecules and proteins. In this paper, we study the problem of learning generative models of graphs from a dataset of graphs of interest. After learning, these models can be used to generate samples with similar properties as the ones in the dataset. Such models can be useful in a lot of applications, e.g. drug discovery and knowledge graph construction. The task of learning generative models of graphs, however, has its unique challenges. In particular, how to handle symmetries in graphs and ordering of its elements during the generation process are important issues. We propose a generic graph neural net based model that is capable of generating any arbitrary graph. We study its performance on a few graph generation tasks compared to baselines that exploit domain knowledge. We discuss potential issues and open problems for such generative models going forward.

## 1 Introduction

Graphs are natural representations of information in many problem domains. For example, relations between entities in knowledge graphs and social networks are well captured by graphs, and they are also good for modeling the physical world, e.g. molecular structure and the interactions between objects in physical systems. Thus, the ability to capture the distribution of a particular family of graphs has many applications. For instance, sampling from the graph model can lead to the discovery of new configurations that share same global properties as is, for example, required in drug discovery (Gómez-Bombarelli et al., 2016). Obtaining graph-structured semantic representations for natural language sentences (Kuhlmann & Oepen, 2016) requires the ability to model (conditional) distributions on graphs. Distributions on graphs can also provide priors for Bayesian structure learning of graphical models (Margaritis, 2003).

Probabilistic models of graphs have been studied for a long time, from at least two perspectives. On one hand, there are random graph models that robustly assign probabilities to large classes of graphs (Erdős & Rényi, 1960; Barabási & Albert, 1999). These make strong independence assumptions and are designed to capture only certain graph properties, like degree distribution and diameter. While these are effective models of the distributions of graphs found in some domains, such as social networks, they are poor models of more richly structured graphs where small structural differences can be functionally significant, such as those encountered in chemistry or when representing the meaning of natural language sentences. As an alternative, a more expressive class of models makes use of graph grammars, which generalize devices from formal language theory so as to produce non-sequential structures (Rozenberg, 1997). Graph grammars are systems of rewrite rules that incrementally derive an output graph via a sequence of transformations of intermediate graphs.While symbolic graph grammars can be made stochastic or otherwise weighted using standard techniques (Droste & Gastin, 2007), from a learnability standpoint, two problems remain. First, inducing grammars from a set of unannotated graphs is nontrivial since formalism-appropriate derivation steps must be inferred and transformed into rules (Lautemann, 1988; Aguiñaga et al., 2016, for example). Second, as with linear output grammars, graph grammars make a hard distinction between what is in the language and what is excluded, making such models problematic for applications where it is inappropriate to assign 0 probability to certain graphs.

In this work we develop an expressive model which makes no assumptions on the graphs and can therefore assign probabilities to any arbitrary graph.[1] Our model generates graphs in a manner similar to graph grammars, where during the course of a derivation new structure (specifically, a new node or a new edge) is added to the existing graph, and where the probability of that addition event depends on the history of the graph derivation. To represent the graph during each step of the derivation, we use a representation based on graph-structured neural networks (graph nets). Recently there has been a surge of interest in graph nets for learning graph representations and solving graph prediction problems (Henaff et al., 2015; Duvenaud et al., 2015; Li et al., 2016; Battaglia et al., 2016; Kipf & Welling, 2016; Gilmer et al., 2017). These models are structured according to the graph being utilized, and are parameterized independent of graph sizes therefore invariant to isomorphism, providing a good match for our purposes.

We evaluate our model by fitting graphs in three problem domains: (1) generating random graphs with certain common topological properties (e.g., cyclicity); (2) generating molecule graphs; and (3) conditional generation of parse trees. Our proposed model performs better than random graph models and LSTM baselines on (1) and (2) and is close to a LSTM sequence to sequence with attention model on (3). We also analyze the challenges our model is facing, e.g. the difficulty of learning and optimization, and discuss possible ways to make it better.

## 2 RELATED WORK

The earliest probabilistic model of graphs developed by Erdős & Rényi (1960) assumed an independent identical probability for each possible edge. This model leads to rich mathematical theory on random graphs, but it is too simplistic to model more complicated graphs that violate this i.i.d. assumption. Most of the more recent random graph models involve some form of "preferential attachment", for example in (Barabási & Albert, 1999) the more connections a node has, the more likely it will be connect to new nodes added to the graph. Another class of graph models aim to capture the small diameter and local clustering properties in graphs, like the small-world model (Watts & Strogatz, 1998). Such models usually just capture one property of the graphs we want to model and are not flexible enough to model a wide range of graphs. Leskovec et al. (2010) proposed the Kronecker graphs model which is capable of modeling multiple properties of graphs, but it still only has limited capacity to allow tractable mathematical analysis.

There are a significant amount of work from the natural language processing and program synthesis communities on modeling the generation of trees. Socher et al. (2011) proposed a recursive neural network model to build parse trees for natural language and visual scenes. Maddison & Tarlow (2014) developed probabilistic models of parsed syntax trees for source code. Vinyals et al. (2015c) flattened a tree into a sequence and then modeled parse tree generation as a sequence to sequence task. Dyer et al. (2016) proposed recurrent neural network models capable of modeling any top-down transition-based parsing process for generating parse trees. Kusner et al. (2017) developed models for context-free grammars for generating SMILES string representations for molecule structures. Such tree models are very good at their task of generating trees, but they are incapable of generating more general graphs that contain more complicated loopy structures.

Our graph generative model is based on a class of neural net models we call graph nets. Originally developed in Scarselli et al. (2009), a range of variants of such graph structured neural net models have been developed and applied to various graph problems more recently (Henaff et al., 2015; Li et al., 2016; Kipf & Welling, 2016; Battaglia et al., 2016; Gilmer et al., 2017). Such models learn representations of graphs, nodes and edges based on a propagation process which communicates information across a graph, and are invariant to graph isomorphism because of the graph size independent parameterization. We use these graph nets to learn representations for making various decisions in the graph generation process.

Our work share some similarity to the recent work of Johnson (2017), where a graph is constructed to solve reasoning problems. The main difference between our work and (Johnson, 2017) is that

---

[1]We may make the analogy to language modeling prior to the advent of RNN language models. On one hand, we had formal grammars that were expressive (e.g., various classes could capture the long range syntactic dependencies found in natural language), but they were brittle and hard to learn; on the other, we had $n$-gram models that were robust and easy to learn, but made naïve Markov assumptions. RNNs offered a way of making models more expressive without increasing fragility or making learning unreasonably difficult.

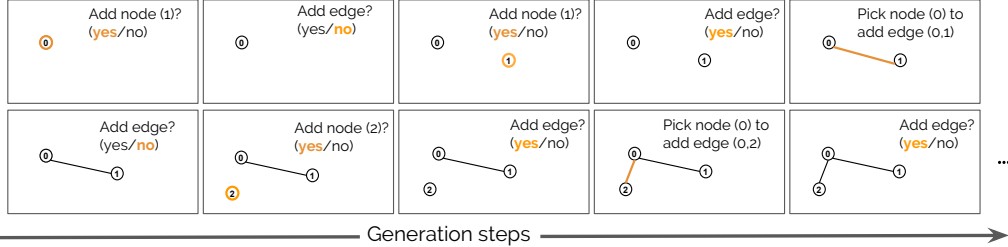

Figure 1: Depiction of the steps taken during the generation process.

our goal in this paper is to learn and represent unconditional or conditional densities on a space of graphs given a representative sample of graphs, whereas Johnson (2017) is primarily interested in using graphs as intermediate representations in reasoning tasks. However, (Johnson, 2017) do offer a probabilistic semantics for their graphs (the soft, real-valued node and connectivity strengths). But, as a generative model, Johnson (2017) did make a few strong assumptions for the generation process, e.g. a fixed number of nodes for each sentence, independent probability for edges given a batch of new nodes, etc.; while our model doesn't make any of these assumptions. On the other side, as we are modeling graph structures, the samples from our model are graphs where an edge or node either exists or does not exist; whereas in (Johnson, 2017) all the graph components, e.g. existence of a node or edge, are all soft, and it is this form of soft node / edge connectivity that was been used for other reasoning tasks. Dense and soft representation may be good for some applications, while the sparse discrete graph structures may be good for others. Potentially, our graph generative model can also be used in an end-to-end pipeline to solve prediction problems as well, like (Johnson, 2017).

## 3 MODEL

Our generative model of graphs is a sequential process which generates one node at a time and connects each node to the partial graph already generated by creating edges one by one.

### 3.1 THE SEQUENTIAL GRAPH GENERATION PROCESS

The actions by which our model generates graphs is illustrated in Figure 1 (for the formal presentation, refer to Algorithm 1 in Appendix A). Briefly, in this generative process, in each iteration we (1) sample whether to add a new node of a particular type or terminate; if a node type is chosen, (2) we add a node of this type to the graph and (3) check if any further edges are needed to connect the new node to the existing graph; if yes (4) we select a node in the graph and add an edge connecting the new node to the selected node. The algorithm goes back to step (3) and repeats until the model decides not to add another edge. Finally, the algorithm goes back to step (1) to add subsequent nodes.

There are many different ways to tweak this generation process. For example, edges can be made directional or typed by jointly modeling the node selection process with type and direction random variables (in the molecule generation experiments below, we use typed nodes and edges). Additionally, constraints on certain structural aspects of graphs can be imposed such as forbidding self-loops or multiple edges between a pair of nodes.

The graph generation process can be seen as a sequence of decisions, i.e., (1) add a new node or not (with probabilities provided by an $f_{addnode}$ module), (2) add a new edge or not (probabilities provided by $f_{addedge}$), and (3) pick one node to connect to the new node (probabilities provided by $f_{nodes}$). One example graph with corresponding decision sequence is shown in Figure 6 in the Appendix. Note that different ordering of the nodes and edges can lead to different decision sequences for the same graph, how to properly handle these orderings is therefore an important issue which we will discuss below.

Once the graph is transformed into such a sequence of structure building actions, we can use a number of different generative models to model it. One obvious choice is to treat the sequences as sentences in natural language, and use conventional LSTM language models. We propose to use graph nets to model this sequential decision process instead. That is, we define the modules that provide probabilities for the structure building events ($f_{addnode}$, $f_{addedge}$ and $f_{nodes}$) in terms of graph nets. As graph nets make use of the structure of the graph to create representations of nodes and

edges via an information propagation process, this parameterization will be more sensitive to the structures being constructed than might be possible in an LSTM-based action sequence model.

## 3.2 Propagation on Graphs and Graph Representations

For any graph $G = (V, E)$, we associate a node embedding vector $\mathbf{h}_v \in \mathbb{R}^H$ with each node $v \in V$. These vectors can be computed initially from node inputs, e.g. node type embeddings, and then propagated on the graph to aggregate information from the local neighborhood. The propagation process is an iterative process, in each round of propagation, a "message" vector is computed on each edge, and after all the messages are computed, each node collects all incoming messages and updates its own representation, as characterized in Eq. 1, 2 and 3, where $f_e$ and $f_n$ are mappings that can be parameterized as neural networks, $\mathbf{x}_{u,v}$ is a feature vector for the edge $(u, v)$, e.g. edge type embedding, $\mathbf{m}_{u \to v}$ is the message vector from $u$ to $v$[2], $\mathbf{a}_v$ is the aggregated incoming message for node $v$ and $\mathbf{h}'_v$ is the new representation for node $v$ after one round of propagation. A typical choice for $f_e$ and $f_n$ is to use fully-connected neural nets for both, but $f_n$ can also be any recurrent neural network core like GRU or LSTM as well. In our experience LSTM and GRU cores perform similarly, we therefore use the simpler GRUs for $f_n$ throughout our experiments.

$$\mathbf{m}_{u \to v} = f_e(\mathbf{h}_u, \mathbf{h}_v, \mathbf{x}_{u,v}) \quad \forall (u, v) \in E, \quad (1)$$

$$\mathbf{a}_v = \sum_{u:(u,v) \in E} \mathbf{m}_{u \to v} \quad \forall v \in V, \quad (2)$$

$$\mathbf{h}'_v = f_n(\mathbf{a}_v, \mathbf{h}_v) \quad \forall v \in V, \quad (3)$$

$$\mathbf{h}_G = \sum_{v \in V} \mathbf{h}_v^G \quad (4)$$

$$\mathbf{h}_G = \sum_{v \in V} \mathbf{g}_v^G \odot \mathbf{h}_v^G \quad (5)$$

Given a set of node embeddings $\mathbf{h}_V = \{\mathbf{h}_1, \ldots, \mathbf{h}_{|V|}\}$, one round of propagation denoted as $\mathrm{prop}(\mathbf{h}_V, G)$ returns a set of transformed node embeddings $\mathbf{h}'_V$ which aggregates information from each node's neighbors (as specified by $G$). It does not change the graph structure. Multiple rounds of propagation, i.e. $\mathrm{prop}(\mathrm{prop}(\cdots(\mathbf{h}_V, G), \cdots, G)$, can be used to aggregate information across a larger neighborhood. Furthermore, different rounds of propagation can have different set of parameters to further increase the capacity of this model, all our experiments use this setting.

To compute a vector representation for the whole graph, we first map the node representations to a higher dimensional $\mathbf{h}_v^G = f_m(\mathbf{h}_v)$, then these mapped vectors are summed together to obtain a single vector $\mathbf{h}_G$ (Eq. 4). The dimensionality of $\mathbf{h}_G$ is chosen to be higher than that of $\mathbf{h}_v$ as the graph contains more information than individual nodes. A particularly useful variant of this aggregation module is to use a separate gating network which predicts $\mathbf{g}_v^G = \sigma(g_m(\mathbf{h}_v))$ for each node, where $\sigma$ is the logistic sigmoid function and $g_m$ is another mapping function, and computes $\mathbf{h}_G$ as a gated sum (Eq. 5). Also the sum can be replaced with other reduce operators like mean or max. We use gated sum in all our experiments. We denote the aggregation operation across the graph without propagation as $\mathbf{h}_G = R(\mathbf{h}_V, G)$.

## 3.3 Probabilities of Structure Building Decisions

Our graph generative model defines a distribution over the sequence of graph generating decisions by defining a probability distribution over possible outcomes for each step. Each of the decision steps is modeled using one of the three modules defined according to the following equations:

$$\mathbf{h}_V^{(T)} = \mathrm{prop}^{(T)}(\mathbf{h}_V, G) \quad (6)$$

$$\mathbf{h}_G = R(\mathbf{h}_V^{(T)}, G) \quad (7)$$

$$f_{addnode}(G) = \mathrm{softmax}(f_{an}(\mathbf{h}_G)) \quad (8)$$

$$f_{addedge}(G, v) = \sigma(f_{ae}(\mathbf{h}_G, \mathbf{h}_v^{(T)})) \quad (9)$$

$$s_u = f_s(\mathbf{h}_u^{(T)}, \mathbf{h}_v^{(T)}), \quad \forall u \in V \quad (10)$$

$$f_{nodes}(G, v) = \mathrm{softmax}(\mathbf{s}) \quad (11)$$

**(a)** $f_{addnode}(G)$ In this module, we take an existing graph $G$ as input, together with its node representations $\mathbf{h}_V$, to produce the parameters necessary to make the decision whether to terminate the algorithm or add another node (this will be probabilities for each node type if nodes are typed).

---

[2]Here we only considered messages along the edge direction $\mathbf{m}_{u \to v}$ for $(u, v) \in E$, but it is also possible to consider the reverse information propagation as well $\mathbf{m}'_{v \to u} = f'_e(\mathbf{h}_u, \mathbf{h}_v)$, and make $\mathbf{a}_v = \sum_{u:(u,v) \in E} \mathbf{m}_{u \to v} + \sum_{u:(v,u) \in E} \mathbf{m}'_{u \to v}$, which is what we used in all experiments.

To compute these probabilities, we first run $T$ rounds of propagation to update node vectors, after which we compute a graph representation vector and predict an output from there through a standard MLP followed by softmax or logistic sigmoid. This process is formulated in Eq. 6, 7, 8. Here the superscript $(T)$ indicates the results after running the propagation $T$ times. $f_{an}$ is a MLP that maps the graph representation vector $\mathbf{h}_G$ to the action output space, here it is the probability (or a vector of probability values) of adding a new node (type) or terminating.

After the predictions are made, the new node vectors $\mathbf{h}_V^{(T)}$ are carried over to the next step, and the same carry-over is applied after each and any decision step. This makes the node vectors recurrent, across both the propagation steps and the different decision steps.

**(b)** $f_{addedge}(G, v)$ This module is similar to (a), we only change the output module slightly as in Eq. 9 to get the probability of adding an edge to the newly created node $v$ through a different MLP $f_{ae}$, after getting the graph representation vector $\mathbf{h}_G$.

**(c)** $f_{nodes}(G, v)$ In this module, after $T$ rounds of propagation, we compute a score for each node (Eq. 10), which is then passed through a softmax to be properly normalized (Eq. 11). $f_s$ maps node state pairs $\mathbf{h}_u$ and $\mathbf{h}_v$ to a score $s_u$ for connecting $u$ to the new node $v$, and $p(\mathbf{y})$ is the output distribution over nodes. This can be extended to handle typed edges by making $s_u$ a vector of scores same size as the number of edge types, and taking the softmax over all nodes and edge types.

**Initializing Node States** Whenever a new node is added to the graph, we need to initialize its state vector. If there are some inputs associated with the node, they can be used to get the initialization vector. We also aggregate across the graph to get a graph vector, and use it as an extra source of input for initialization. More concretely, the node state for a new node $v$ is initialized as the following:

$$\mathbf{h}_v = f_{init}(R_{init}(\mathbf{h}_V, G), \mathbf{x}_v). \tag{12}$$

Here $\mathbf{x}_v$ is any input feature associated with the node, e.g. node type embeddings, and $R_{init}(\mathbf{h}_V, G)$ computes a graph representation, $f_{init}$ is an MLP. If not using $R_{init}(\mathbf{h}_V, G)$ as part of the input to the initialization module, nodes with the same input features added at different stages of the generation process will have the same initialization. Adding the graph vector fixes this issue.

**Conditional Generative Model** The graph generative model described above can also be used to do conditional generation, where some input is used to condition the generation process. We only need to make a few minor changes to the model architecture, by making a few design decisions about where to add in the conditioning information.

The conditioning information comes in the form of a vector, and then it can be added in one or more of the following modules: (1) the propagation process; (2) the output component for the three modules, i.e. in $f_n$, $f_e$ and $f_s$; (3) the node state initialization module $f_{init}$. In our experiments, we use the conditioning information only in $f_n$ and $f_{init}$. Standard techniques for improving conditioning like attention can also be used, where we can use the graph representation to compute a query vector.

## 3.4 Training and Evaluation

Our graph generative model defines a joint distribution $p(G, \pi)$ over graphs $G$ and node and edge ordering $\pi$ (corresponding to the derivation in a traditional graph grammar). When generating samples, both the graph itself and an ordering are generated by the model. For both training and evaluation, we are interested in the marginal $p(G) = \sum_{\pi \in \mathcal{P}(G)} p(G, \pi)$. This marginal is, however, intractable to compute for moderately large graphs as it involves a sum over all possible permutations. To evaluate this marginal likelihood we therefore need to use either sampling or some approximation instead. One Monte-Carlo estimate is based on importance sampling, where

$$p(G) = \sum_\pi p(G, \pi) = \sum_\pi q(\pi \mid G) \frac{p(G, \pi)}{q(\pi \mid G)} = \mathbb{E}_{q(\pi|G)} \left[ \frac{p(G, \pi)}{q(\pi \mid G)} \right]. \tag{13}$$

Here $q(\pi|G)$ is any proposal distribution over permutations, and the estimate can be obtained by generating a few samples from $q(\pi \mid G)$ and then average $p(G, \pi)/q(\pi \mid G)$ for the samples. The variance of this estimate is minimized when $q(\pi \mid G) = p(\pi \mid G)$. When a fixed canonical ordering is available for any arbitrary $G$, we can use it to train and evaluate our model by taking $q(\pi \mid G)$ to be a delta function that puts all the probability on this canonical ordering. This choice of $q$, however,

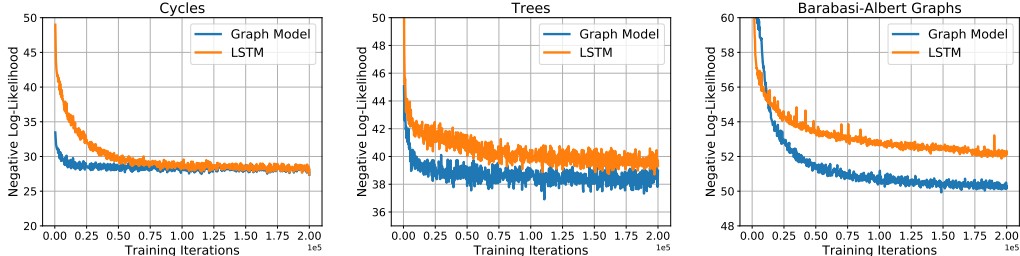

Figure 2: Training curves for the graph model and LSTM model on three sets.

only gives us a lower bound on the true marginal likelihood as it does not have full support over the set of all permutations.

In training, since direct optimization of $\log p(G)$ is intractable, we can therefore learn the joint distribution $p(G, \pi)$ instead by maximizing the expected joint log-likelihood

$$\mathbb{E}_{p_{data}(G,\pi)}[\log p(G, \pi)] = \mathbb{E}_{p_{data}(G)}\mathbb{E}_{p_{data}(\pi|G)}[\log p(G, \pi)]. \tag{14}$$

Given a dataset of graphs, we can get samples from $p_{data}(G)$ fairly easily, and we have the freedom to choose $p_{data}(\pi|G)$ for training. Since the maximizer of Eq. 14 is $p(G, \pi) = p_{data}(G, \pi)$, to make the training process match the evaluation process, we can take $p_{data}(\pi \mid G) = q(\pi \mid G)$. Training with such a $p_{data}(\pi \mid G)$ will drive the posterior of the model distribution $p(\pi \mid G)$ close to the proposal distribution $q(\pi \mid G)$, therefore improving the quality of our estimate of the marginal probability.

Ordering is an important issue for our graph model, in the experiments we always use a fixed ordering or uniform random ordering for training, and leave the potentially better solution of learning an ordering to future work. In particular, in the learning to rank literature there is an extensive body of work on learning distributions over permutations, for example the Mallows model (Mallows, 1957) and the Plackett-Luce model (Plackett, 1975; Luce, 1959), which may be used here. Interested readers can also refer to (Leskovec et al., 2010; Vinyals et al., 2015a; Stewart et al., 2016) for discussions of similar ordering issues from different angles.

## 4 EXPERIMENTS

We study the properties and performance of different graph generation models and odering strategies on three different tasks. More experiment results and detailed settings are included in Appendix C.

### 4.1 GENERATION OF GRAPHS WITH CERTAIN TOPOLOGICAL PROPERTIES

In the first experiment, we train graph generative models on three sets of synthetic undirected graphs: (1) cycles, (2) trees, and (3) graphs generated by the Barabasi–Albert model (Barabási & Albert, 1999), which is a good model for power-law degree distribution. We generate data on the fly during training, all cycles and trees have between 10 to 20 nodes, and the Barabasi–Albert model is set to generate graphs of 15 nodes and each node is connected to 2 existing nodes when added to the graph.

For comparison, we contast our model against the Erdős & Rényi (1960) random graph model and a LSTM baseline. We estimate the edge probability parameter $p$ in the Erdős–Rényi model using maximum likelihood. For the LSTM model, we sequentialized the decision sequences (see Figure 6 for an example) used by the graph model and trained LSTM language models on them. During training, for each graph we uniformly randomly permute the orderings of the nodes and order the edges by node indices, and then present the permuted graph to the graph model and the LSTM model. In experiments on all three sets, we used a graph model with node state dimensionality of 16 and set the number of propagation steps $T = 2$, and the LSTM model has a hidden state size of 64. The two models have roughly the same number of parameters (LSTM 36k, graph model 32k).

The training curves plotting $-\log p(G, \pi)$ with $G, \pi$ sampled from the training distribution, comparing the graph model and the LSTM model, are shown in Figure 2. From these curves we can clearly see that the graph models train faster and have better asymptotic performance as well.

| Dataset | Graph Model | LSTM | Erdős–Rényi Model |
|---|---|---|---|
| Cycles | **84.4%** | 48.5% | 0.0% |
| Trees | **96.6%** | 30.2% | 0.3% |
| Barabasi–Albert Graphs | **0.0013** | 0.0537 | 0.3715 |

Table 1: Percentage of valid samples for three models on cycles and trees datasets, and the KL-divergence between the degree distributions of samples and data for Barabasi–Albert graphs.

Since our graphs have topological properties, we can also evaluate the samples of these models and see how well they align with these properties. We generated 10,000 samples from each model. For cycles and trees, we evaluate what percentage of samples are actually cycles or trees. For graphs generated by the Barabasi–Albert model, we compute the node degree distribution. The results are shown in Table 1 and Figure 3. Again we can see that the proposed graph model has the capability of matching the training data well in terms of all these metrics. Note that we used the same graph model on three different sets of graphs, and the model learns to adapt to the data.

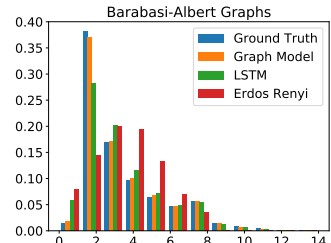

Figure 3: Degree histogram for samples generated by models trained on Barabasi–Albert Graphs. The histogram labeled "Ground Truth" shows the data distribution estimated from 10,000 examples.

Here the success of the graph model compared to the LSTM baseline can be partly attributed to the ability to refer to specific nodes in a graph. The ability to do this inevitably requires keeping track of a varying set of objects and then pointing to them, which is non-trivial for a LSTM to do. Pointer networks (Vinyals et al., 2015b) can be used to handle the pointers, but building a varying set of objects is challenging in the first place, and the graph model provides a way to do it.

## 4.2 MOLECULE GENERATION

In the second experiment, we train graph generative models for the task of molecule generation. Recently, there has been a number of papers tackling this problem by using RNN language models on SMILES string representations of molecules (Gómez-Bombarelli et al., 2016; Segler et al., 2017; Bjerrum & Threlfall, 2017). An example of a molecule and its corresponding SMILES string are shown in Figure 4. Kusner et al. (2017) took one step further and used context free grammar to model the SMILES strings. However, inherently molecules are graph structured objects where it is possible to have cycles.

Figure 4: `NNc1nncc(O)n1`

We used the ChEMBL database (the latest version, 23) for this study; previous versions of ChEMBL were also used in (Segler et al., 2017; Olivecrona et al., 2017) for molecule generation. We filtered the database and chose to model molecules with at most 20 heavy atoms. This resulted in a training / validation / testing split of 130,830 / 26,166 / 104,664 examples each. The chemical toolkit RDKit (2006) is used to convert between the SMILES strings and the graph representation of the molecules. Both the nodes and the edges in molecule graphs are typed. All the model hyperparameters are tuned on the validation set, number of propagation steps $T$ is chosen from $\{1, 2\}$.

We compare the graph model with baseline LSTM language models trained on the SMILES strings as well as the graph generating sequences used by the graph model. RDKit can produce canonical SMILES representations for each molecule with associated edge ordering, we therefore train the models using these canonicalized representations. We also trained these models with permuted ordering. For the graph model, we randomly permute the node ordering and change the edge ordering correspondingly, for the LSTM on SMILES, we first convert the SMILES string into a graph representation, permute the node ordering and then convert back to a SMILES string without canonicalization, similar to (Bjerrum, 2017).

| Model | Gen.Seq | Ordering | $N$ | NLL | %valid | %valid and novel |
|-------|---------|----------|-----|-----|--------|------------------|
| LSTM | SMILES | Fixed | 1 | 21.48 | 93.59 | 81.27 |
| LSTM | SMILES | Random | $< 100$ | **19.99** | 93.48 | 83.95 |
| LSTM | Graph | Fixed | 1 | 22.06 | 85.16 | 80.14 |
| LSTM | Graph | Random | $O(n!)$ | 63.25 | 91.44 | 91.26 |
| Graph | Graph | Fixed | 1 | 20.55 | **97.52** | 90.01 |
| Graph | Graph | Random | $O(n!)$ | 58.36 | 95.98 | **95.54** |

Table 2: Results on the molecule generation task. $N$ is the number of permutations for each molecule the model is trained on. Typically the number of different SMILES strings for each molecule $< 100$.

| Model | Gen.Seq | Ordering | $N$ | Fixed Ordering | Best Ordering | Marginal |
|-------|---------|----------|-----|----------------|---------------|----------|
| LSTM | SMILES | Fixed | 1 | 17.28 | 15.98 | 15.90 |
| LSTM | SMILES | Random | $< 100$ | **15.95** | 15.76 | 15.67 |
| LSTM | Graph | Fixed | 1 | 16.79 | 16.35 | 16.26 |
| LSTM | Graph | Random | $O(n!)$ | 20.57 | 18.90 | 15.96 |
| Graph | Graph | Fixed | 1 | 16.19 | **15.75** | 15.64 |
| Graph | Graph | Random | $O(n!)$ | 20.18 | 18.56 | **15.32** |

Table 3: Negative log-likelihood evaluation on small molecules with no more than 6 nodes.

We evaluate the negative log-likelihood for all models with the canonical ordering on the test set. We also generate 100,000 samples from each model and evaluate how many of them are valid well-formatted molecule representations and how many of the generated samples are not already seen in the training set following (Segler et al., 2017; Olivecrona et al., 2017). The results are shown in Table 2, which also lists the type of graph generating sequence and the ordering the models are trained on. Note that the models trained with random ordering are not tailored to the canonical ordering used in evaluation. In Appendix C.2, we show the distribution of a few chemical metrics for the generated samples to further assess the their quality. The LSTM on SMILES strings has a slight edge in terms of likelihood evaluated under canonical ordering (which is domain specific), but the graph model generates significantly more valid and novel samples. It is also interesting that the LSTM model trained with random ordering improves performance on canonical ordering, this is probably related to overfitting. Lastly, when compared using the generic graph generation decision sequence, the Graph architecture outperforms LSTM in NLL as well.

It is intractable to estimate the marginal likelihood $p(G) = \sum_\pi p(G, \pi)$ for large molecules. However, for small molecules this is possible. We did the enumeration and evaluated the 6 models on small molecules with no more than 6 nodes. As we evaluate, we compare the negative log-likelihood we got with the fixed ordering and the best possible ordering, as well as the true marginal, the results are shown in Table 3. On these small molecules, the graph model trained with random ordering has better marginal likelihood, and surprisingly for the models trained with fixed ordering, the canonical ordering they are trained on are not always the best ordering, which suggests that there are big potential for actually learning an ordering.

Figure 5 shows a visualization of the molecule generation processes for the graph model. The model trained with canonical ordering learns to generate nodes and immediately connect it to the latest part of the generated graph, while the model trained with random ordering took a completely different approach by generating pieces first and then connect them together at the end.

## 4.3 Parse Tree Generation

In the last experiment, we look at a conditional graph generation task - generating parse trees given an input natural language sentence. We took the Wall Street Journal dataset with sequentialized parse trees used in (Vinyals et al., 2015c), and trained LSTM sequence to sequence models with attention as the baselines on both the sequentialized trees as well as on the decision sequences used by the graph model. In the dataset the parse trees are sequentialized following a top-down depth-first traversal ordering, we therefore used this ordering to train our graph model as well. Besides this, we also conducted experiments using the breadth-first traversal ordering. We changed our graph model slightly and replaced the loop for generating edges to a single step that picks one node as the parent

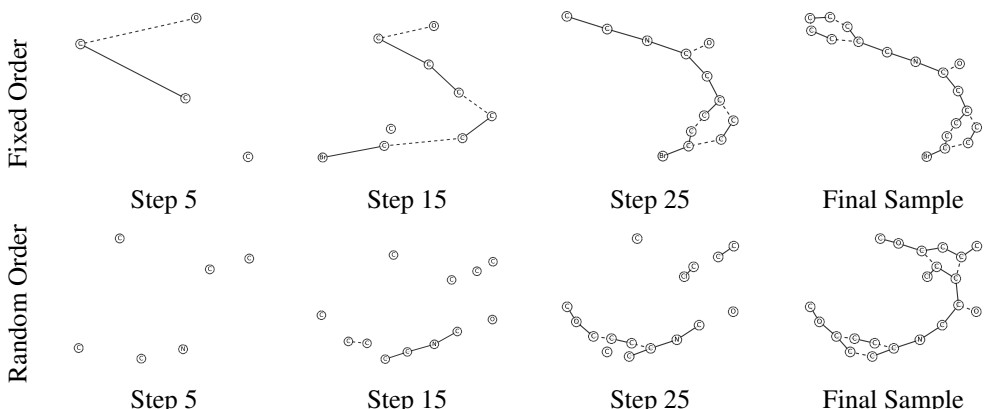

Figure 5: Visualization of the molecule generation processes for graph model trained with fixed and random ordering. Solid lines represent single bonds, and dashed lines represent double bounds.

| Model | Gen.Seq | Ordering | Perplexity | %Correct |
|-------|---------|----------|------------|----------|
| LSTM | Sequentialized Tree | Depth-First | **1.114** | **31.1** |
| LSTM | Sequentialized Tree | Breadth-First | 1.187 | 28.3 |
| LSTM | Graph | Depth-First | 1.158 | 26.2 |
| LSTM | Graph | Breadth-First | 1.399 | 0.0 |
| Graph | Graph | Depth-First | 1.124 | 28.7 |
| Graph | Graph | Breadth-First | 1.238 | 21.5 |

Table 4: Parse tree generation results, evaluated on the Eval set.

for each new node to adapt to the tree structure. This shortens the decision sequence for the graph model, although the flattened parse tree sequence the LSTM uses is still shorter. We also employed an attention mechanism to get better conditioning information as for the sequence to sequence model.

Table 4 shows the perplexity results of different models on this task. Since the length of the decision sequences for the graph model and sequentialized trees are different, we normalized the log-likelihood of all models using the length of the flattened parse trees to make them comparable. To measure sample quality we used another metric that checks if the generated parse tree exactly matches the ground truth tree. From these results we can see that the LSTM on sequentialized trees is better on both metrics, but the graph model does better than the LSTM trained on the same and more generic graph generating decision sequences, which is compatible with what we observed in the molecule generation experiment.

One important issue for the graph model is that it relies on the propagation process to communicate information on the graph structure, and during training we only run propagation for a fixed $T$ steps, and in this case $T = 2$. Therefore after a change to the tree structure, it is not possible for other remote parts to be aware of this change in such a small number of propagation steps. Increasing $T$ can make information flow further on the graph, however the more propagation steps we use the slower the graph model would become, and more difficult it would be to train them. For this task, a tree-structured model like R3NN (Parisotto et al., 2016) may be a better fit which can propagate information on the whole tree by doing one bottom-up and one top-down pass in each iteration. On the other hand, the graph model is modeling a longer sequence than the sequentialized tree sequence, and the graph structure is constantly changing therefore so as the model structure, which makes training of such graph models to be considerably harder than LSTMs.

## 5 DISCUSSIONS AND FUTURE DIRECTIONS

The graph model in the proposed form is a powerful model capable of generating arbitrary graphs. However, as we have seen in the experiments and the analysis, there are still a number of challenges facing these models. Here we discuss a few of these challenges and possible solutions going forward.

**Ordering** Ordering of nodes and edges is critical for both learning and evaluation. In the experiments we always used predefined distribution over orderings. However, it may be possible to learn an ordering of nodes and edges by treating the ordering $\pi$ as a latent variable, this is an interesting direction to explore in the future.

**Long Sequences** The generation process used by the graph model is typically a long sequence of decisions. If other forms of sequentializing the graph is available, e.g. SMILES strings or flattened parse trees, then such sequences are typically 2-3x shorter. This is a significant disadvantage for the graph model, it not only makes it harder to get the likelihood right, but also makes training more difficult. To alleviate this problem we can tweak the graph model to be more tied to the problem domain, and reduce multiple decision steps and loops to single steps.

**Scalability** Scalability is a challenge to the graph generative model we proposed in this paper. Large graphs typically lead to very long graph generating sequences. On the other side, the graph nets use a fixed $T$ propagation steps to propagate information on the graph. However, large graphs require large $T$s to have sufficient information flow, this would also limit the scalability of these models. To solve this problem, we may use models that sequentially sweep over edges, like (Parisotto et al., 2016), or come up with ways to do coarse-to-fine generation.

**Difficulty in Training** We have found that training such graph models is more difficult than training typical LSTM models. The sequence these models are trained on are really long, but also the model structure is constantly changing, which leads to various scaling issues and only adds to the difficulty. We found lowering the learning rate can solve a lot of the instability problem, but more satisfying solutions may be obtained by tweaking the model architecture.

## 6 CONCLUSION

In this paper, we proposed a powerful deep generative model capable of generating arbitrary graphs through a sequential process. We studied its properties on a few graph generation problems. This model has shown great promise and has unique advantages over standard LSTM models. We hope that our results can spur further research in this direction to obtain better generative models of graphs.

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

---

**Algorithm 1** Generative Process for Graphs

---

1: $E_0 = \phi, V_0 = \phi, G_0 = (V_0, E_0), t = 1$        ▷ Initial graph is empty
2: $\mathbf{p}_t^{addnode} \leftarrow f_{addnode}(G_{t-1})$        ▷ Probabilities of initial node type and STOP
3: $v_t \sim \text{Categorical}(\mathbf{p}_t^{addnode})$        ▷ Sample initial node type or STOP
4: **while** $v_t \neq \text{STOP}$ **do**
5:      $V_t \leftarrow V_{t-1} \cup \{v_t\}$        ▷ Incorporate node $v_t$
6:      $E_{t,0} \leftarrow E_{t-1}, i \leftarrow 1$
7:      $p_{t,i}^{addedge} \leftarrow f_{addedge}((V_t, E_{t,0}), v_t)$        ▷ Probability of adding an edge to $v_t$
8:      $z_{t,i} \sim \text{Bernoulli}(p_{t,i}^{addedge})$        ▷ Sample whether to add an edge to $v_t$
9:      **while** $z_{t,i} = 1$ **do**        ▷ Add edges pointing to new node $v_t$
10:          $\mathbf{p}_{t,i}^{nodes} \leftarrow f_{nodes}((V_t, E_{t,i-1}), v_t)$        ▷ Probabilities of selecting each node in $V_t$
11:          $v_{t,i} \sim \text{Categorical}(\mathbf{p}_{t,i}^{nodes})$
12:          $E_{t,i} \leftarrow E_{t,i-1} \cup \{(v_{t,i}, v_t)\}$        ▷ Incorporate edge $v_t - v_{t,i}$
13:          $i \leftarrow i + 1$
14:          $p_{t,i}^{addedge} \leftarrow f_{addedge}((V_t, E_{t,i-1}), v_t)$        ▷ Probability of adding another edge
15:          $z_{t,i} \sim \text{Bernoulli}(p_{t,i}^{addedge})$        ▷ Sample whether to add another edge to $v_t$
16:      **end while**
17:      $E_t \leftarrow E_{t,i-1}$
18:      $G_t \leftarrow (V_t, E_t)$
19:      $t \leftarrow t + 1$
20:      $\mathbf{p}_t^{addnode} \leftarrow f_{addnode}(G_{t-1})$        ▷ Probabilities of each node type and STOP for next node
21:      $v_t \sim \text{Categorical}(\mathbf{p}_t^{addnode})$        ▷ Sample next node type or STOP
22: **end while**
23: **return** $G_t$

---

## A    GRAPH GENERATION PROCESS

The graph generation process is presented in Algorithm 1 for reference.

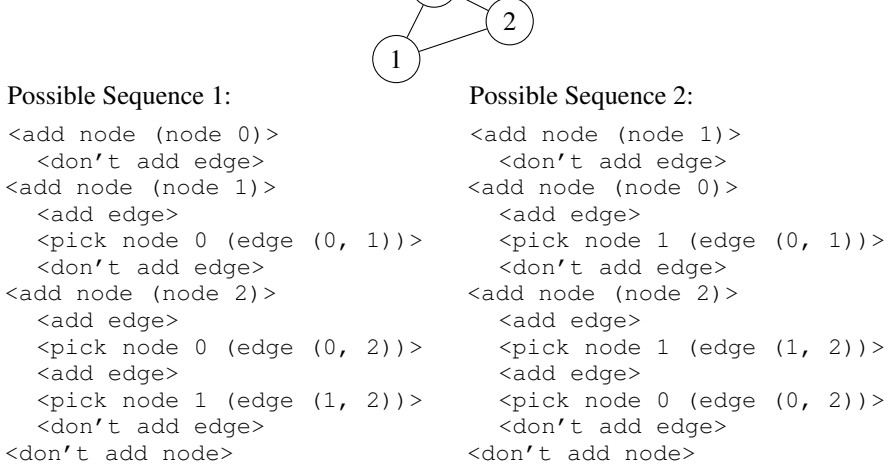

Possible Sequence 1:
```
<add node (node 0)>
  <don't add edge>
<add node (node 1)>
  <add edge>
  <pick node 0 (edge (0, 1))>
  <don't add edge>
<add node (node 2)>
  <add edge>
  <pick node 0 (edge (0, 2))>
  <add edge>
  <pick node 1 (edge (1, 2))>
  <don't add edge>
<don't add node>
```

Possible Sequence 2:
```
<add node (node 1)>
  <don't add edge>
<add node (node 0)>
  <add edge>
  <pick node 1 (edge (0, 1))>
  <don't add edge>
<add node (node 2)>
  <add edge>
  <pick node 1 (edge (1, 2))>
  <add edge>
  <pick node 0 (edge (0, 2))>
  <don't add edge>
<don't add node>
```

Figure 6: An example graph and two corresponding decision sequences.

Figure 6 shows an example graph. Here the graph contains three nodes $\{0, 1, 2\}$, and three edges $\{(0, 1), (0, 2), (1, 2)\}$. Consider generating nodes in the order of 0, 1 and 2, and generating edge $(0, 2)$ before $(1, 2)$, then the corresponding decision sequence is the one shown on the left. Here the decisions are indented to clearly show the two loop levels. On the right we show another possible generating sequence generating node 1 first, and then node 0 and 2. In general, for each graph there might be many different possible orderings that can generate it.

## B   MODEL IMPLEMENTATION DETAILS

In this section we present more implementation details about our graph generative model.

### B.1   THE PROPAGATION MODEL

The message function $f_e$ is implemented as a fully connected neural network, as the following:
$$\mathbf{m}_{u \to v} = f_e(\mathbf{h}_u, \mathbf{h}_v, \mathbf{x}_{u,v}) = \mathrm{MLP}(\mathrm{concat}([\mathbf{h}_u, \mathbf{h}_v, \mathbf{x}_{u,v}])).$$
We can also use an additional edge function $f_e'$ to compute the message in the reverse direction as
$$\mathbf{m}_{v \to u}' = f_e'(\mathbf{h}_u, \mathbf{h}_v, \mathbf{x}_{u,v}) = \mathrm{MLP}'(\mathrm{concat}([\mathbf{h}_u, \mathbf{h}_v, \mathbf{x}_{u,v}])).$$
When not using reverse messages, the node activation vectors are computed as
$$\mathbf{a}_v = \sum_{u:(u,v)\in E} \mathbf{m}_{u \to v}.$$
When reverse messages are used, the node activations are
$$\mathbf{a}_v = \sum_{u:(u,v)\in E} \mathbf{m}_{u \to v} + \sum_{u:(v,u)\in E} \mathbf{m}_{u \to v}'.$$
The node update function $f_n$ is implemented as a recurrent cell in RNNs, as the following:
$$\mathbf{h}_v' = \mathrm{RNNCell}(\mathbf{h}_v, \mathbf{a}_v),$$
where RNNCell can be a vanilla RNN cell, where
$$\mathbf{h}_v' = \sigma(\mathbf{W}\mathbf{h}_v + \mathbf{U}\mathbf{a}_v),$$
a GRU cell
$$\mathbf{z}_v = \sigma(\mathbf{W}_z \mathbf{h}_v + \mathbf{U}_z \mathbf{a}_v),$$
$$\mathbf{r}_v = \sigma(\mathbf{W}_r \mathbf{h}_v + \mathbf{U}_z \mathbf{a}_v),$$
$$\tilde{\mathbf{h}}_v = \tanh(\mathbf{W}(\mathbf{r}_v \odot \mathbf{h}_v) + \mathbf{U}\mathbf{a}_v),$$
$$\mathbf{h}_v' = (1 - \mathbf{z}_v) \odot \mathbf{h}_v + \mathbf{z}_v \odot \tilde{\mathbf{h}}_v,$$
or an LSTM cell
$$\mathbf{i}_v = \sigma(\mathbf{W}_i \mathbf{h}_v + \mathbf{U}_i \mathbf{a}_v + \mathbf{V}_i \mathbf{c}_v),$$
$$\mathbf{f}_v = \sigma(\mathbf{W}_f \mathbf{h}_v + \mathbf{U}_f \mathbf{a}_v + \mathbf{V}_v \mathbf{c}_v),$$
$$\tilde{\mathbf{c}}_v = \tanh(\mathbf{W}_c \mathbf{h}_v + \mathbf{U}_c \mathbf{a}_v),$$
$$\mathbf{c}_v' = \mathbf{f}_v \odot \mathbf{c}_v + \mathbf{i}_v \odot \tilde{\mathbf{c}}_v,$$
$$\mathbf{o}_v' = \sigma(\mathbf{W}_o \mathbf{h}_v + \mathbf{U}_o \mathbf{a}_v + \mathbf{V}_o \mathbf{c}_v'),$$
$$\mathbf{h}_v' = \mathbf{o}_v' \odot \tanh(\mathbf{c}_v').$$

In the experiments, we used a linear layer in the message functions $f_e$ in place of the MLP, and we set the dimensionality of the outputs to be twice the dimensionality of the node state vectors $\mathbf{h}_u$. For the synthetic graphs and molecules, $f_e$ and $f_e'$ share the same set of parameters, while for the parsing task, $f_e$ and $f_e'$ have different parameters. We always use GRU cells in our model. Overall GRU cells and LSTM cells perform equally well, and both are significantly better than the vanilla RNN cells, but GRU cells are slightly faster than the LSTM cells.

Note that each round of propagation can be thought of as a graph propagation "layer". When propagating for a fixed number of $T$ rounds, we can have tied parameters on all layers, but we found using different parameters on all layers perform consistently better. We use untied weights in all experiments.

For aggregating across the graph to get graph representation vectors, we first map the node representations $\mathbf{h}_v$ into a higher dimensional space $\mathbf{h}_v^G = f_m(\mathbf{h}_v)$, where $f_m$ is another MLP, and then $\mathbf{h}_G = \sum_{v \in V} \mathbf{h}_v^G$ is the graph representation vector. We found gated sum
$$\mathbf{h}_G = \sum_{v \in V} \mathbf{g}_v^G \odot \mathbf{h}_v^G$$
to be consistently better than a simple sum, where $\mathbf{g}_v^G = \sigma(g_m(\mathbf{h}_v))$ is a gating vector. In the experiments we always use this form of gated sum, and both $f_m$ and $g_m$ are implemented as a single linear layer, and the dimensionality of $\mathbf{h}_G$ is set to twice the dimensionality of $\mathbf{h}_v$.

## B.2 THE OUTPUT MODEL

**(a)** $f_{addnode}(G)$    This module takes an existing graph as input and produce a binary (non-typed nodes) or categorical output (typed nodes). More concretely, after obtaining a graph representation $\mathbf{h}_G$, we feed that into an MLP $f_{an}$ to output scores. For graphs where the nodes are not typed, we have $f_{an}(\mathbf{h}_G) \in \mathbb{R}$ and the probability of adding one more node is

$$f_{addnode}(G) = p(\text{add one more node}|G) = \sigma(f_{an}(\mathbf{h}_G)).$$

For graphs where the nodes can be one of $K$ types, we make $f_{an}$ output a $K+1$-dimensional vector $f_{an}(\mathbf{h}_G) \in \mathbb{R}^{K+1}$, and

$$\hat{\mathbf{p}} = [\hat{p}_1, ..., \hat{p}_{K+1}]^\top = f_{an}(\mathbf{h}_G)$$

$$p_k = \frac{\exp(\hat{p}_k)}{\sum_{k'} \exp(\hat{p}'_k)}, \qquad \forall k$$

then

$$p(\text{add one more node with type } k|G) = p_k.$$

We add an extra type $K+1$ to represent the decision of not adding any more nodes.

In the experiments, $f_{an}$ is always implemented as a linear layer and we found this to be sufficient.

**(b)** $f_{addedge}(G, v)$    This module takes the current graph and a newly added node $v$ as input and produces a probability of adding an edge. In terms of implementation it is treated as exactly the same as (a), except that we add the new node into the graph first, and use a different set of parameters both in the propagation module and in the output module where we use a separate $f_{ae}$ in place of $f_{an}$. This module always produces Bernoulli probabilities, i.e. probability for either adding one edge or not. Typed edges are handled in (c).

**(c)** $f_{nodes}(G, v)$    This module picks one of the nodes in the graph to be connected to node $v$. After propagation, we have node representation vectors $\mathbf{h}_u^{(T)}$ for all $u \in V$, then a score $s_u \in \mathbb{R}$ for each node $u$ is computed as

$$s_u = f_s(\mathbf{h}_u^{(T)}, \mathbf{h}_v^{(T)}) = \text{MLP}(\text{concat}([\mathbf{h}_u^{(T)}, \mathbf{h}_v^{(T)}])),$$

The probability of a node being selected is then a softmax over these scores

$$p_u = \frac{\exp(s_u)}{\sum_{u'} \exp(s_{u'})}.$$

For graphs with $J$ types of edges, we produce a vector $s_u \in \mathbb{R}^J$ for each node $u$, by simply changing the output size of the MLP for $f_s$. Then the probability of a node $u$ and edge type $j$ being selected is a softmax over all scores across all nodes and edge types

$$p_{u,j} = \frac{\exp(s_{u,j})}{\sum_{u',j'} \exp(s_{u',j'})}.$$

## B.3 INITIALIZATION AND CONDITIONING

When a new node $v$ is created, its node vector $\mathbf{h}_v$ need to be initialized. In our model the node vector $\mathbf{h}_v$ is initialized using inputs from a few different sources: (1) a node type embedding or any other node features that are available; (2) a summary of the current graph, computed as a graph representation vector after aggregation; (3) any conditioning information, if available.

Among these, (1) node type embedding $\mathbf{e}$ comes from a standard embedding module; (2) is implemented as a graph aggregation operation, more specifically

$$\mathbf{h}_G^{init} = \sum_{v \in V} \mathbf{g}_v^{init} \odot \mathbf{h}_v^{init}$$

where $\mathbf{g}_v^{init}$ and $\mathbf{h}_v^{init}$ are the gating vectors and projected node state vectors as described in B.1, but with different set of parameters; (3) is a conditioning vector $\mathbf{c}$ if available.

$\mathbf{h}_v$ is then initialized as

$$\mathbf{h}_v = f_{init}(\mathbf{e}, \mathbf{h}_G^{init}, \mathbf{c}) = \text{MLP}(\text{concat}([\mathbf{e}, \mathbf{h}_G^{init}, \mathbf{c}])).$$

The conditioning vector $\mathbf{c}$ summarizes any conditional input information, for images this can be the output of a convolutional neural network, for text this can be the output of an LSTM encoder. In the parse tree generation task, we employed an attention mechanism similar to the one used in Vinyals et al. (2015c).

More specifically, we used an LSTM to obtain the representation of each input word $\mathbf{h}_i^c$, for $i \in \{1, ..., L\}$. Whenever a node is created in the graph, we compute a query vector

$$\mathbf{h}_G^q = \sum_{v \in V} \mathbf{g}_v^q \odot \mathbf{h}_v^q$$

which is again an aggregate over all node vectors. This query vector is used to compute a score for each input word as

$$u_i^c = v^\top \tanh(\mathbf{W}\mathbf{h}_i^c + \mathbf{U}\mathbf{h}_G^q),$$

these scores are transformed into weights

$$\mathbf{a}^c = \text{Softmax}(\mathbf{u}^c),$$

where $\mathbf{a}^c = [a_1^c, ..., a_L^c]^\top$ and $\mathbf{u}^c = [u_1^c, ..., u_L^c]^\top$. The conditioning vector $\mathbf{c}$ is computed as

$$\mathbf{c} = \sum_i a_i^c \mathbf{h}_i^c.$$

### B.4 LEARNING

For learning we have a set of training graphs, and we train our model to maximize the expected joint likelihood $\mathbb{E}_{p_{data}(G)} \mathbb{E}_{p_{data}(\pi|G)} [\log p(G, \pi)]$ as discussed in Section 3.4.

Given a graph $G$ and a specified ordering $\pi$ of the nodes and edges, we can obtain a particular graph generating sequence (Appendix A shows an example of this). The log-likelihood $\log p(G, \pi)$ can then be computed for this sequence, where the likelihood for each individual step is computed using the output modules described in B.2.

For $p_{data}(\pi|G)$ we explored two possibilities: (1) canonical ordering in the particular domain; (2) uniform random ordering. The canonical ordering is a fixed ordering of a graph nodes and edges given a graph. For molecules, the SMILES string specified an ordering of nodes and edges which we use as the canonical ordering. In the implementation we used the default ordering provided in the chemical toolbox rdkit as the canonical ordering. For parsing we tried two canonical orderings, depth-first-traversal ordering and breadth-first-traversal ordering. For uniform random ordering we first generate a random permutation of node indices which gives us the node ordering, and then sort the edges according to the node indices to get edge ordering. When evaluating the marginals we take the permutations on edges into account as well.

## C MORE EXPERIMENT DETAILS AND RESULTS

In this section we describe more detailed experiment setup and present more experiment results not included in the main paper.

### C.1 SYNTHETIC GRAPH GENERATION

For this experiment the hidden size of the LSTM model is set to 64 and the size of node states in the graph model is 16, number of propagation steps $T = 2$.

For both models we selected the learning rates from $\{0.001, 0.0005, 0.0002\}$ on each of the three sets. We used the Adam (Kingma & Ba, 2014) optimizer for both.

## C.2 Molecule Generation

**Model Details** Our graph model has a node state dimensionality of 128, the LSTM models have hidden size of 512. The two models have roughly the same number of parameters (around 2 million). Our graph model uses GRU cores as $f_n$, we have tried LSTMs as well but they perform similarly as GRUs. We have also tried GRUs for the baselines, but LSTM models work slightly better. The node state dimensionality and learning rate are chosen according to grid search in $\{32, 64, 128, 256\} \times \{0.001, 0.0005, 0.0002, 0.0001\}$, while for the LSTM models the hidden size and learning rate are chosen from $\{128, 256, 512, 1024\} \times \{0.001, 0.0005, 0.0002\}$. The best learning rate for the graph model is $0.0001$, while for the LSTM model the learning rate is $0.0002$ or $0.0005$. The LSTM model used a dropout rate of 0.5, while the graph model used a dropout rate of 0.2 which is applied to the last layer of the output modules. As discussed in the main paper, the graph model is significantly more unstable than the LSTM model, and therefore a much smaller learning rate should be used. The number of propagation steps $T$ is chosen from $\{1, 2\}$, increasing $T$ is in principle beneficial for the graph representations, but it is also more expensive. For this task a small $T$ is already showing a good performance so we didn't explore much further. Overall the graph model is roughly 2-3x slower than the LSTM model with similar amount of parameters in our comparison.

**Distribution of chemical properties for samples** Here we examine the distribution of chemical metrics for the valid samples generated from trained models. For this study we chose a range of chemical metrics available from RDKit (2006), and computed the metrics for 100,000 samples generated from each model. For reference, we also computed the same metrics for the training set, and compare the sample metrics with the training set metrics.

For each metric, we create a histogram to show its distribution across the samples, and compare the histogram to the histogram on the training set by computing the KL divergence between them. The results are shown in Figure 7. Note that all models are able to match the training distribution on these metrics quite well, notably the graph model and LSTM model trained on permuted node and edge sequences has a bias towards generating molecules with higher SA scores which is a measure of the ease of synthesizing the molecules. This is probably due to the fact that these models are trained to generate molecular graphs in arbitrary order (as apposed to following the canonical order that makes sense chemically), therefore more likely to generate things that are harder to synthesize. However, this can be overcome if we train with RL to optimize for this metric. The graph model trained with permuted nodes and edges also has a slight bias toward generating larger molecules with more atoms and bonds.

We also note that the graph and LSTM models trained on permuted nodes and edge sequences can still be improved as they are not even overfitting after 1 million training steps. This is because with node and edge permutation, these models see on the order of $n!$ times more data than the other models. Given more training time these models can improve further.

**Changing the bias for $f_{addnode}$ and $f_{addedge}$** Since our graph generation model is very modular, it is possible to tweak the model after it has been trained. For example, we can tweak a single bias parameter in $f_{addnode}$ and $f_{addedge}$ to increase or decrease the graph size and edge density.

In Figure 8 (a) we show the shift in the distribution of number of atoms for the samples when changing the $f_{addnode}$ bias. As the bias changes, the samples change accordingly while the model is still able to generate a high percentage of valid samples.

Figure 8 (b) shows the shift in the distribution of number of bonds for the samples when changing the $f_{addedge}$ bias. The number of bonds, i.e. number of edges in the molecular graph, changes as this bias changes. Note that this level of fine-grained control of edge density in sample generation is not straightforward to achieve with LSTM models trained on SMILES strings. Note that however here the increasing the $f_{addedge}$ slightly changed the average node degree, but negatively affected the total number of bonds. This is because the edge density also affected the molecule size, and when the bias is negative, the model tend to generate larger molecules to compensate for this change, and when this bias is positive, the model tend to generate smaller molecules. Combining $f_{addedge}$ bias and $f_{addnode}$ bias can achieve the net effect of changing edge density.

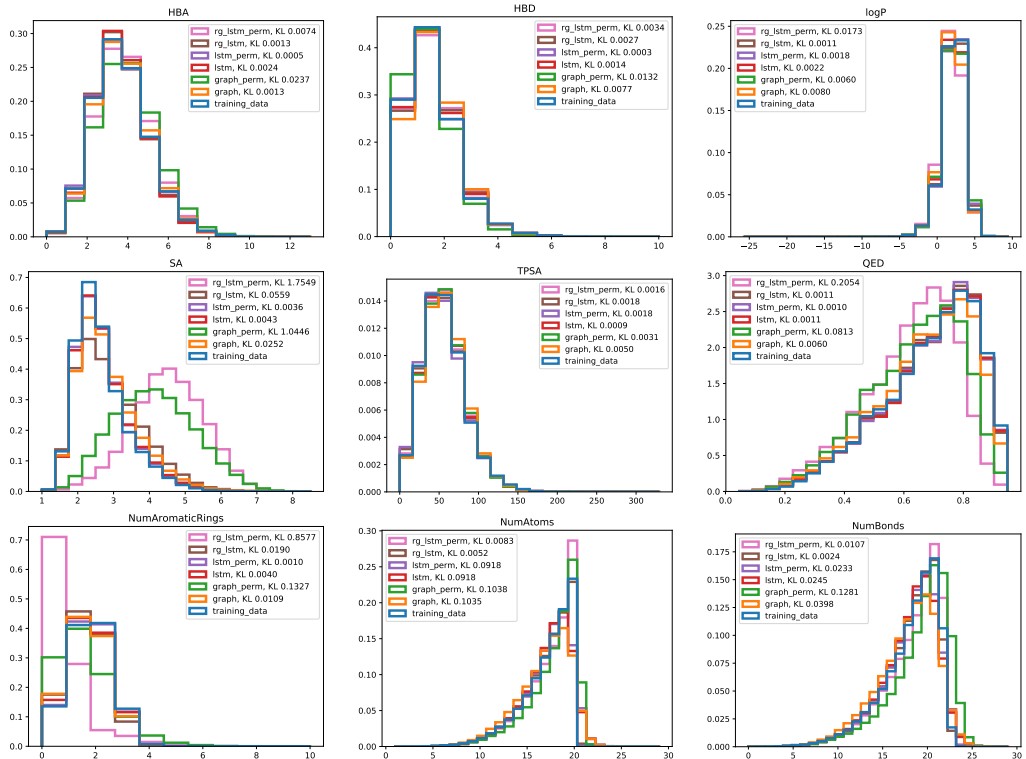

Figure 7: Distribution of chemical properties for samples from different models and the training set. *rg_lstm*: LSTM trained on fixed graph generation decision sequence; *rg_lstm_perm*: LSTM trained on permuted graph generation decision sequence; *lstm*: LSTM on SMILES strings; *lstm_perm*: LSTM on SMILES strings with permuted nodes; *graph*: graph model on fixed node and edge sequence; *graph_perm*: graph model on permuted node and edge sequences.

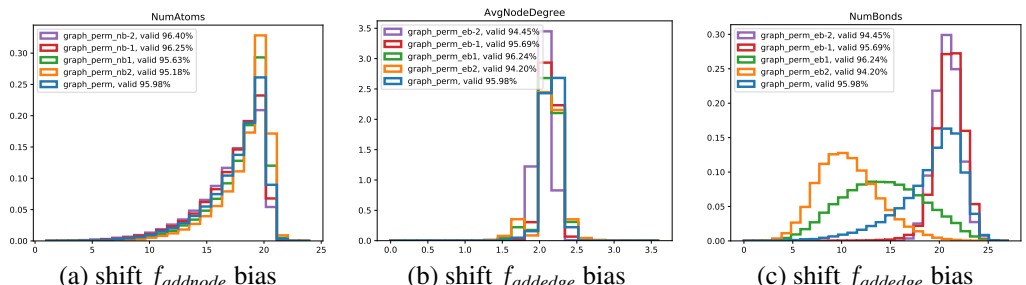

(a) shift $f_{addnode}$ bias      (b) shift $f_{addedge}$ bias      (c) shift $f_{addedge}$ bias

Figure 8: Changing the $f_{addnode}$ and $f_{addedge}$ biases can affect the generated samples accordingly, therefore achieving a level of fine-grained control of sample generation process. *nb<bias>* and *eb<bias>* shows the bias values added to the logits.

**Step-by-step molecule generation visualization**      Here we show a few examples for step-by-step molecule generation. Figure 9 shows an example of such step-by-step generation process for a graph model trained on canonical ordering, and Figure 10 shows one such example for a graph model trained on permuted random ordering.

**Overfitting the Canonical Ordering**      When trained with canonical ordering, our model will adapt its graph generating behavior to the ordering it is being trained on, Figure 9 and Figure 10 show examples on how the ordering used for training can affect the graph generation behavior.

On the other side, training with canonical ordering can result in overfitting more quickly than training with uniform random ordering. In our experiments, training with uniform random ordering rarely

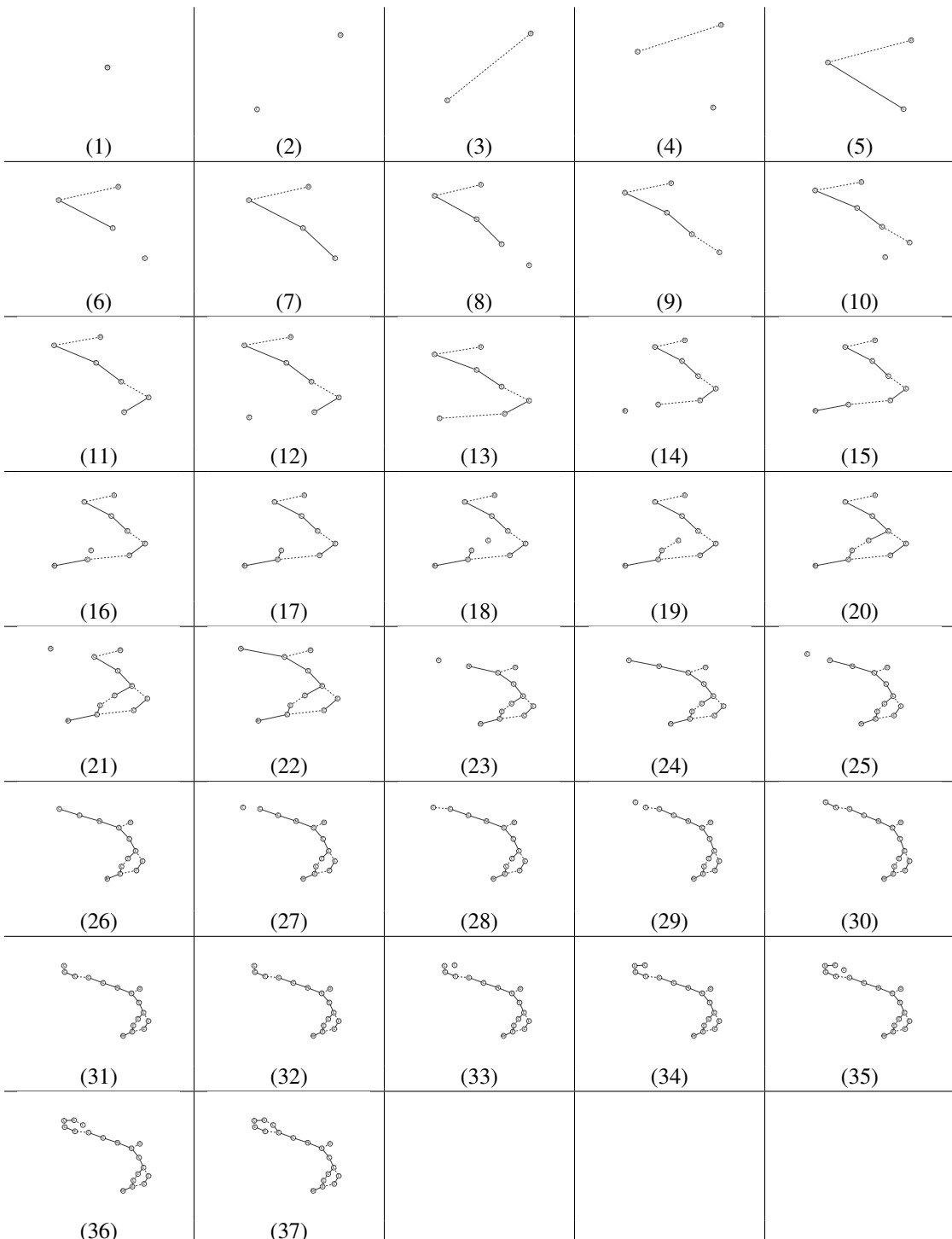

Figure 9: Step-by-step generation process visualization for a graph model trained with canonical ordering.

overfits at all, but with canonical ordering the model overfits much more quickly. Effectively, with random ordering the model will see potentially factorially many possible orderings for the same graph, which can help reduce overfitting, but this also makes learning harder as many orderings do not exploit the structure of the graphs at all.

Another interesting observation we have about training with canonical ordering is that models trained with canonical ordering may not assign the highest probabilities to the canonical ordering after training. From Table 3 we can see that the log-likelihood results for the canonical ordering (labeled "fixed ordering") is not always the same as the best possible ordering, even though they are quite close.

Figure 11 shows an example histogram of negative log-likelihood $\log p(G, \pi)$ across all possible orderings $\pi$ for a small molecule under a model trained with canonical ordering. We can see that the small negative log-likelihood values concentrate on very few orderings, and a large number of orderings have significantly larger NLL. This shows that the model can learn to concentrate probabilities to orderings close to the canonical ordering, but it still "leaks" some probability to other orderings.

### C.3    PARSING TASK

**Model Details**    In this experiment we used a graph model with node state dimensionality of 64, and an LSTM encoder with hidden size 256. Attention over input is implemented using a graph aggregation operation to compute a query vector and then use it to attend to the encoder LSTM states, as described in B.3. The baseline LSTM models have hidden size 512 for both the encoder and the decoder. Dropout of 0.5 is applied to both the encoder and the decoder. For the graph model the dropout in the decoder is reduced to 0.2 and applied to various output modules and the node initialization module. The baseline models have more than 2 times more parameters than the graph model (52M vs 24M), mostly due to using a larger encoder.

The node state dimensionality for the graph model and the hidden size of the encoder LSTM is chosen from a grid search $\{16, 32, 64, 128\} \times \{128, 256, 512\}$. For the LSTM seq2seq model the size of the encoder and decoder are always tied and selected from $\{128, 256, 512\}$. For all models the learning rate is selected from $\{0.001, 0.0005, 0.0002\}$.

For the LSTM encoder, the input text is always reversed, which empirically is silghtly better than the normal order.

For the graph model we experimented with $T \in \{1, 2, 3, 4, 5\}$. Larger $T$ can in principle be beneficial for getting better graph representations, however this also means more computation time and more instability. $T = 2$ results in a reasonable balance for this task.

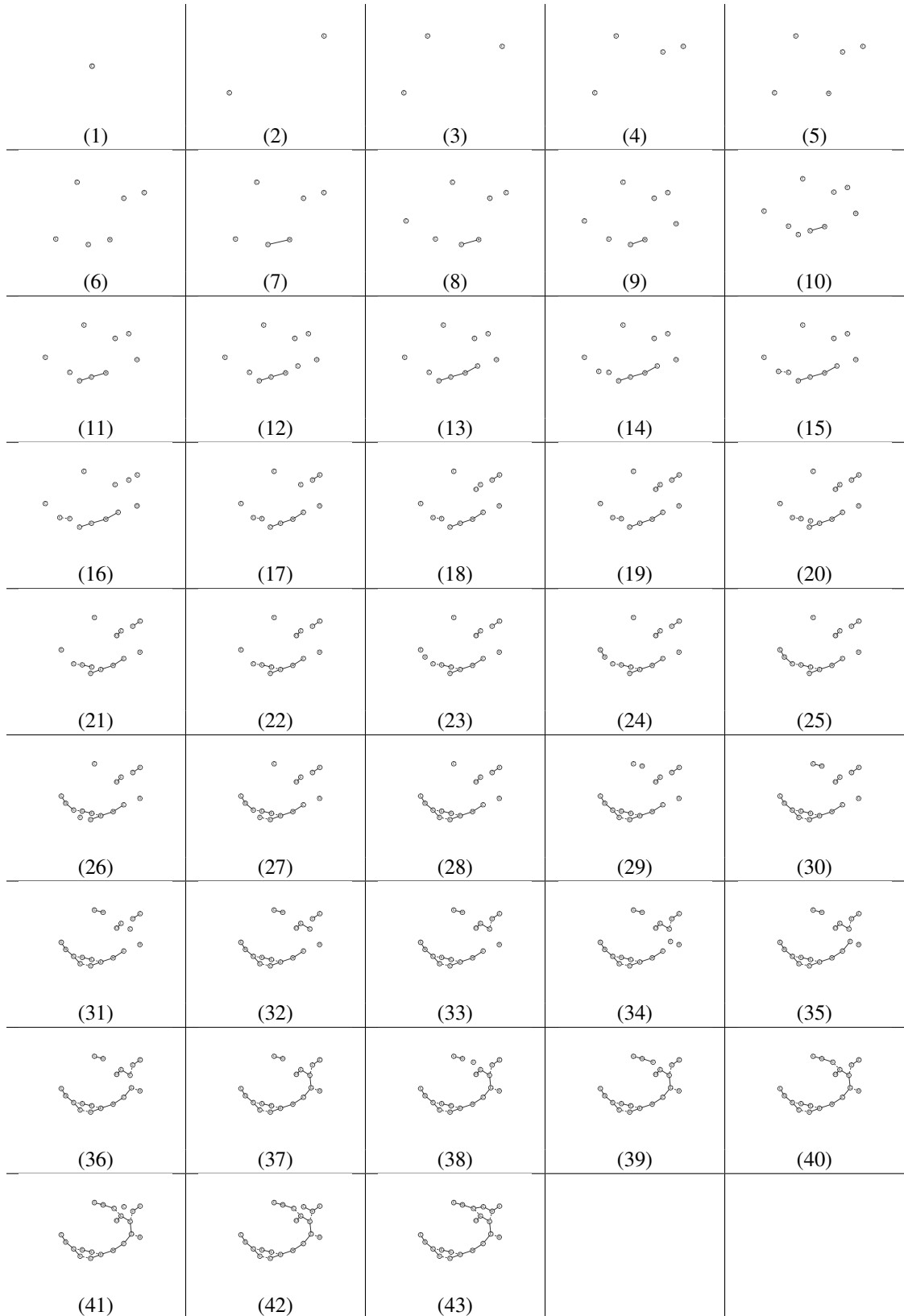

Figure 10: Step-by-step generation process visualization for a graph model trained with permuted random ordering.

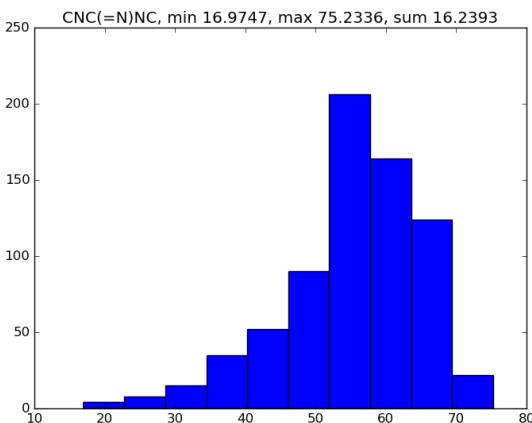

Figure 11: Histogram of negative log-likelihood $\log p(G, \pi)$ under different orderings $\pi$ for one small molecule under a model trained with canonical ordering.

