# OpenReview forum: "Learning Deep Generative Models of Graphs"
_ICLR.cc/2018/Conference — Invite to Workshop Track_

### Official Review · AnonReviewer2 · 2017-11-27
**Learning deep generative models for graphs**

**Rating:** 5
**Confidence:** 3

**Review:**

The paper introduces a generative model for graphs. The three main decision functions in the sequential process are computed with neural nets. The neural nets also compute node embeddings and graph embeddings and the embeddings of the current graph are used to compute the decisions at time step T. The paper is well written but, in my opinion, a description of the learning framework should be given in the paper. Also, a summary of the hyperparameters used in the proposed system should be given. It is claimed that all possible types of graphs can be learned which seems rather optimistic. For instance, when learning trees, the system is tweaked for generating trees. Also, it is not clear whether models for large graphs can be learned. The paper contain many interesting contributions but, in my opinion, the model is too general and the focus should be given on some retricted classes of graphs. Therefore, I am not convinced that the paper is ready for publication at ICLR'18.

* Introduction. I am not convinced by the discussion on graph grammars in the second paragraph. It is known that there does not exist a definition of regular grammars in graph (see Courcelle and Engelfriet, graph structure and monadic second-order logic ...). Moreover, many problems are known to be undecidable. For weighted automata, the reference Droste and Gastin considers weighted word automata and weighted logic for words. Therefore I does not seem pertinent here. A more complete reference is "handbook of weighted automata" by Droste. Also, many decision problems for wighted automata are known to be undecidable. I am not sure that the paragraph is useful for the paper. A discussion on learning as in footnote 1 shoud me more interesting.
* Related work. I am not expert in the field but I think that there are recent references which could be cited for probablistic models of graphs.
* Section 3.1. Constraints can be introduced to impose structural properties of the generated graphs. This leads to the question of cheating in the learning process.
* Section 3.2. The functions f_m and g_m for defining graph embedding are left undefined. As the graph embedding is used in the generating process and for learning, the functions must be defined and their choice explained and justified.
* Section 3. As said before, a general description of the learning framework should be given. Also, it is not clear to me how the node and graph embeddings are initialized and how they evolve along the learning process. Therefore, it is not clear to me why the proposed updating framework for the embeddings allow to generate decision functions adapted to the graphs to be learned.  Consequently, it is difficult to see the influence of T. Also, it should be said whether the node embeddings and graph embeddings for the output graph can be useful.
* Section 3. A summary of all the hyperparameters should be given.
* Section 4.1. The number of steps is not given. Do you present the same graph multiple times. Why T=2 and not 1 or 10 ?
* Section 4.2. From table 2, it seems that all permutations are used for training which is rather large for molecules of size 20. Do you use tweaks in the generation process.
* Section 4.3. The generation process is adapted for generating trees which seems to be cheating. Again the choice of T seems ad hoc and based on computational burden.
* Section 5 should contain a discussion on complexity issues because it is not clear how the model can learn large graphs.
* Section 5. The discussion on the difficulty of training shoud be emphasized and connected to the --missing-- description of the model architecture and its hyperparameters.
* acronyms should be expansed at their first use

---

> ### Comment · AnonReviewer2 · 2018-01-09
> **Rebuttal response**
>
> Thanks to the authors for the rebuttal and their modifications. In my opinion, the problem of generating large graphs remains. Also, the interplay between the intermediate node/graph representations and the generation process during learning remains unclear to me.

---

### Official Review · AnonReviewer3 · 2017-11-27
**A promising generative model for graphs**

**Rating:** 6
**Confidence:** 3

**Review:**

The authors proposed a graph neural network based architecture for learning generative models of graphs. Compared with traditional learners such as LSTM, the model is better at capturing graph structures and provides a flexible solution for training with arbitrary graph data. The representation is clear with detailed empirical studies. I support its acceptance.

The draft does need some improvements and here is my suggestions.
1. Figure 1 could be improved using a concrete example like in Figure 6. If space allowed, an example of different ordering leads to the same graph will also help.

2. More details on how node embedding vectors are initialized. How does different initializations affect results? Why is nodes at different stages with the same initialization problematic?

3. More details of how conditioning information is used, especially for the attention mechanism used later in parse tree generation.

4. The sequence ordering is important. While the draft avoids the issue theoretically, it does has interesting results in molecule generation experiment. I suggest the authors at least discuss the empirical over-fitting problem with respect to ordering.

5. In Section 4.1, the choice of ER random graph as a baseline is too simplistic. It does not provide a meaningful comparison. A better generative model for cycles and trees could help.

6. When comparing training curves with LSTM, it might be helpful to also include the complexity comparison of each iteration.

---

### Official Review · AnonReviewer4 · 2017-12-01
**Revisiting auto-regressive models for graph generation**

**Rating:** 6
**Confidence:** 4

**Review:**

The authors introduce a sequential/recurrent model for generation of small graphs. The recurrent model takes the form of a graph neural network. Similar to RNN language models, new symbols (nodes/edges) are sampled from Bernoulli or categorical distributions which are parameterized by small fully-connected neural networks conditioned on the last recurrent hidden state.

The paper is very well written, nicely structured, provides extensive experimental evaluation, and examines an important problem that has so far not received much attention in the field.

The proposed model has several interesting novelties (mainly in terms of new applications/experiments, and being fully auto-regressive), yet also shares many similarities with the generative component of the model introduced in [1] (not cited): Both models make use of (recurrent) graph neural networks to learn intermediate node representations, from which they predict whether new nodes/edges should be added or not. [1] speeds this process up by predicting multiple nodes and edges at once, whereas in this paper, such a multi-step process is left for future work. Training the generative model with fixed ground-truth ordering was similarly performed in [1] (“strong supervision”) and is thus not particularly novel.

Eqs.1-3: Why use recurrent formulation in both the graph propagation model and in the auto-regressive main loop (h_v -> h_v’)? Have the authors experimented with other variants (dropping the weight sharing in either or both of these steps)?

Ordering problem: A solution for the ordering problem was proposed in [2]: learning a matching function between the orderings of model output and ground truth. A short discussion of this result would make the paper stronger.

For chemical molecule generation, a direct comparison to some more recent work (e.g. the generator of the grammar VAE [3]) would be insightful.

Other minor points:
- In the definition of f_nodes: What is p(y)? It would be good to explicitly state that (boldface) s is a vector of scores s_u (or score vectors, in case of multiple edge types) for all u in V.
- The following statement is unclear to me: “but building a varying set of objects is challenging in the first place, and the graph model provides a way to do it.” Maybe this can be substantiated by experimental results (e.g. a comparison against Pointer Networks [4])?
- Typos in this sentence: “Lastly, when compared using the genaric graph generation decision sequence, the Graph architecture outperforms LSTM in NLL as well.”

Overall I feel that this paper can be accepted with some revisions (as discussed above), as, even though it shares many similarities with previous work on a very related problem, it is well-written, well-presented and addresses an important problem.

[1] D.D. Johnson, Learning Graphical State Transitions, ICLR 2017
[2] R. Stewart, M. Andriluka, and A. Y. Ng, End-to-End People Detection in Crowded Scenes, CVPR 2016
[3] M.J. Kusner, B. Paige, J.M. Hernandez-Lobato, Grammar Variational Autoencoder, ICML 2017
[4] O. Vinyals, M. Fortunato, N. Jaitly, Pointer Networks, NIPS 2015

---

> ### Comment · AnonReviewer4 · 2018-01-02
> **Rebuttal response**
>
> I would like to thank the authors for their detailed response and for adding a model description section in appendix A that clarifies implementation details.
>
> As pointed out in my initial review, I still feel that the paper misses a direct experimental comparison against some related established work, which is why I am not willing to change my review score at this point. As mentioned in my review, I think it would be best to compare (or at least comment on why such a comparison was left out) against work such as the Grammar VAE (M.J. Kusner, B. Paige, J.M. Hernandez-Lobato, Grammar Variational Autoencoder, ICML 2017).

---

> > ### Author Response · Authors · 2018-01-05
> > **We have added a new comment w.r.t. the grammar VAE**
> >
> > Thank you for your review and for suggesting a comparison against grammar VAE.
> >
> > We have tried grammar VAE on our dataset.  Please see our latest comment above for more detail.

---

> > > ### Comment · AnonReviewer4 · 2018-01-12
> > > **Grammar VAE comparison**
> > >
> > > Thanks for adding this comparison against the Grammar VAE model. I think it certainly allows for a better placement of your proposed model w.r.t related work.
> > >
> > > While I hoped this comparison would tell a clear story about whether a) decoding with a grammar or b) decoding directly into a graph representation would work better, it seems that the results that you report raise a few more questions.
> > >
> > > It is unclear to me why your proposed LSTM decoder baseline (without respecting SMILES grammar) should work so much better (in terms of number of valid molecules) than the recurrent decoder of the grammar VAE. You mention that you suspect that the lack of full auto-regressiveness (output is not fed back as input to the next time step) might be a distinguishing factor. Since there is such a significant difference in model performance, this would certainly have to be experimentally verified in order to be an acceptable explanation for this difference. What happens if your LSTM decoder is trained with teacher forcing instead (like the original SMILES VAE paper https://arxiv.org/pdf/1610.02415v1.pdf) or without any new input at each time step? Will it similarly degrade performance and explain the difference?
> > >
> > > All in all, I stick to my original evaluation of the paper as I think the paper offers a promising approach for generating (small) graphs which certainly deserves attention by the community. The experimental evaluation is extensive while some points (see above) still require clarification. I hope the authors can address my last few questions should the paper be accepted (or for some later later venue).

---

### Public Comment · (anonymous) · 2017-11-14
**Relation to Learning Graphical State Transitions**

I've enjoyed reading this paper, but I'm wondering if the authors are aware of "Learning Graphical State Transitions" (Johnson, ICLR'17 oral). The work presented here feels like a generalization, but it shares many ideas with the earlier paper, and a discussion of the differences would definitely be very helpful.

---

> ### Author Response · Authors · 2017-11-16
> **Reply**
>
> Thanks for the comment and bringing up this related paper.  We will update our paper with more discussion and citations to related work (we are not allowed to make changes to our submission at the moment).
>
> The main difference between our work and Johnson (2017) is that our goal in this paper is to learn and represent unconditional or conditional densities on a space of graphs given a representative sample of graphs, whereas Johnson is primarily interested in using graphs as intermediate representations in reasoning tasks.  However, Johnson (2017) do offer a probabilistic semantics for their graphs (the soft, real-valued node and connectivity strengths).  But, as a generative model, Johnson (2017) did make a few strong assumptions for the generation process, e.g. a fixed number of nodes for each sentence, independent probability for edges given a batch of new nodes, etc.; while our model doesn't make any of these assumptions.
>
> On the other side, as we are modeling graph structures, the samples from our model are graphs where an edge or node either exists or does not exist; whereas in Johnson (2017) all the graph components, e.g. existence of a node or edge, are all soft, and it is this form of soft node / edge connectivity that was been used for other reasoning tasks.  Dense and soft representation may be good for some applications, while the sparse discrete graph structures may be good for others.  Potentially, our graph generative model can also be used in an end-to-end pipeline to solve prediction problems as well, like Johnson (2017).

---

### Author Response · Authors · 2017-12-19
**We thank the reviewers for the reviews and have updated the paper**

We thank the reviewers for the thoughtful reviews and suggestions, and for recognizing the significance and novelty of this work.  Graph generation is an important topic, and our work provides a generic framework that is capable of generating arbitrary graphs through learning from data.

We have updated the submission to address some of the comments we received so far, which help us making this paper better, in particular:

- We have added an entire section B in the appendix to describe model implementation details which should help clarifying confusions, as all reviewers raised this concern.  In addition we have added detailed hyperparameter settings for all tasks in appendix C to make the results more reproducible.

- We added a reference to “Learning Graph State Transitions” [1] and discussed the relationship and differences between our work and [1] at the end of section 2.  As reviewer 4 and the anonymous comment pointed out, both our work and [1] share some similarities.  However [1] mostly uses a graph as intermediate representation to help solving reasoning tasks, while we aim to learn an unconditional or conditional probabilistic model of a distribution of graphs from a sample of representative graphs.  As generative models of graphs, [1] assigns soft strengths for each node and edge, while in our generative model in a sample a node / edge either exists or does not exist.  [1] also made a few strong assumptions about the graph generation process, while we don’t make any such assumptions.  See the paper for more details.

- We added a sentence at the end of the paragraph following equations (1)-(3) to address Reviewer 4’s comments on weight sharing, explaining that the parameters in different rounds of propagation don’t have to be tied, and in the experiments we always use different parameters in different propagation rounds which empirically is consistently better than tied weights.  Reviewer 4 also suggested we may drop weight sharing in the outer recurrent loop as well, but this is hard as the graph generating sequences are not fixed length sequences.  It is unclear how dropping weight sharing would work here.

- We added some extra discussion on learning an ordering to the last paragraph of section 3.4.  We thank Reviewer 4 for pointing out the related work of [2].  [2] described a way to match a set of ground truths to a sequence of candidates generated by a model, which is related to learning an ordering.  We have added this reference in the paper.  Applying such a matching-based solution seems challenging in our setting though, as it is unclear how this can be used to learn a distribution over graphs, and we don’t have a clear distance metric between the generated graph components and the reference graph.  Learning such a distance metric by itself seems to be a nontrivial task.  We are aware of some other literature on learning an ordering / permutation, in particular from the learning to rank community, we have added a few other references in this direction and hope this can provide some alternative insights on this problem.

- We modified figure 1 and added another possible graph generating sequence to figure 6, which Reviewer 3 suggested could make the presentation clearer.

- We added a paragraph discussing the effect of fitting the fixed canonical ordering to the end of Appendix C.2.  As reviewer 3 pointed out, our model may overfit to a particular ordering if it is always trained with that ordering.  In the experiments we do observe that the model assigns higher probability to the canonical ordering it is being trained on, and much less probability to other orderings.  However, in some cases the canonical ordering does not have the highest probability under a trained model, as can be also seen from Table 3, where the likelihood under fixed ordering (the ordering being trained on) is not always the same as the likelihood under the best possible ordering.  This indicates there may be potential in learning an ordering improving the canonical one.

- We changed the paragraph on the “dependence on T” to focus more on the challenges w.r.t. scalability, as Reviewer 2 mentioned this could make the paper clearer.

- We changed a few numbers in Table 2 and 3 to reflect our latest results after the deadline, which does not change the overall conclusion on the comparison between different approaches.

We will try to add more experimental results to the paper when they are ready.

[1] Learning Graph State Transitions.  Daniel D Johnson.  ICLR 2017.
[2] R. Stewart, M. Andriluka, and A. Y. Ng, End-to-End People Detection in Crowded Scenes.  CVPR 2016

---

> ### Author Response · Authors · 2017-12-19
> **More comments on other concerns raised in the reviews**
>
> In the following we clarify a few other concerns raised by the reviewers:
>
> Reviewer 4: comparison against Pointer Networks
>
> Pointer networks provide a way to select and output items from a set of candidates.  We used this pointer-style mechanism in our model in the node selection module f_nodes.  Standard pointer nets assume a set of candidates is given, e.g. the input sequence as a set of candidate tokens in a seq2seq framework.  In our model we learn to construct this set of candidates (a set of nodes in the graph) starting from an empty set, which is non-trivial.
>
> Reviewer 2: model is too general, better focus on restricted classes of graphs
>
> The primary goal of this work is to have a powerful generic model capable of generating any arbitrary graphs.  This is an important but not well-studied task as recognized by Reviewers 3 and 4.  For long we have specific models designed for restricted classes of graphs, e.g. models of trees, and models that capture some properties of graphs like the random graph models discussed in the related work, but to our knowledge our work is the first generic model that is capable of generating any type of graphs.  The model is powerful and can adapt its graph generating behavior by learning from data.  Comparing our proposed model to the previous graph generative models is in spirit analogous to the contrast between RNN language models and grammar-based or n-gram language models.
>
> Reviewer 2: the model is tweaked for generating trees, this seems to be cheating
>
> In the experiments in section 4.1, we used the exact same model to learn on three different datasets without any tweaking for each individual dataset, learning to generate cycles, trees and Barabasi-Albert graphs, and our proposed model can successfully adapt and generate graphs similar to each of these three datasets.  In the parsing experiment in section 4.3, we removed the inner loop and always generate one edge fore each new node.  This simplified the model and introduced a bit more structure into our model, which results in a performance improvement.   Note that the baselines we compared against also exploits the tree structure, in particular the sequentialized trees encode the tree structure with opening and closing brackets, which is very effective, and this information is not available to the graph model as we trained exclusively on the very generic graph generating sequences.
>
> Reviewer 2: discussion on graph grammars not clear
>
> The questions about decidability in graph grammars is a mostly orthogonal to the point of the paper. We included this discussion to provide context to the paper since graph grammars (of various classes) and automata have been widely used in attempts to formalize generative models of graphs. Corcelle’s undecidability/impossibility results are precisely why we are taking an alternative approach to modeling graphs in this work.

---

### Author Response · Authors · 2018-01-05
**New results comparing against the grammar VAE baseline**

We have tried the grammar VAE on our dataset and did a comparison with the results reported in our paper.  The experiment was based on the published code for grammar VAE available here: https://github.com/mkusner/grammarVAE

The code did not work directly on our data, so we made a few tweaks:
- The grammar they used was tailored to their dataset, and did not directly work on our dataset, i.e. many of our molecules cannot be generated by their grammar.  We have added a few more grammar rules to make it work on our dataset.
- They used a variant of the VAE loss where the weighting of the reconstruction part and the KL part of the loss did not directly correspond to the standard ELBO bound, so we changed it to make the VAE bound comparable to our likelihood estimates.
- No sampling code was provided in the codebase, so we added our own implementation based on their code.  The sampling process generates random latents from the prior N(0,1) and then pass them through the decoder with all the grammar handling described in the Algorithm 1 in the grammar VAE paper, this part was used for evaluation.

After training, the grammar VAE achieves a negative ELBO of 11.98 on the test set, but out of 100,000 samples generated from the trained model, only 29.56% are valid SMILES strings.  Note the 11.98 ELBO bound is considerably better than reported in our paper with the best numbers around 20, but the fraction of valid SMILES strings is a lot worse than our results where it is easy to get over 90% valid, but this result is on par with the reported numbers in the grammar VAE paper where around 31(+/- 7) % are valid after Bayesian optimization.

These results are a bit surprising, we try to interpret these results with the following explanations:
- Our graph model and the LSTM baseline are capable of modeling a wider class of molecules than the grammar VAE due to the limited capability of the grammar.  The grammar used in grammar VAE is a context-free grammar with a set of simple expansion rules, which is enough for modeling our datasets.  But our models would still assign some probability to more complicated graphs (those with nested loops for example) therefore leading to a lower likelihood number.
- The grammar offers very strong domain knowledge that is very helpful for shaping the likelihood of a given string.  In the implementation the grammar is used to zero-out inapplicable expansion rules and renormalize the rest which can significantly boost the likelihood of a given sequence where our model and the LSTM baseline do not have access to any of these.
- However, when sampling, the grammar is still quite brittle as it can generate many invalid strings, for example unpaired digits for rings, and invalid valence for certain atoms.  To capture these more complex behaviors more complicated grammars need to be used, which requires significant expert knowledge.  Our approach and the LSTM baseline does not use any such domain knowledge.  In our evaluation the quality of the generated samples from the grammar VAE model is considerably worse than both our model and the LSTM baseline.
- The decoder of the grammar VAE model is not fully auto-regressive, i.e. the output of one step is not fed back to the model as the input to the next step, making it fully auto-regressive may improve performance.

Overall, our approach offers a very generic and powerful solution to the graph generation problem without the need of domain expertise, while the grammar VAEs went the opposite route which relies on expert knowledge (the grammar).  Nevertheless, we can combine our graph generation model with domain knowledge including grammars to help us in the graph generation process to further improve performance.

---

### Decision · Program_Chairs · 2018-01-29
**ICLR 2018 Conference Acceptance Decision**

**Decision:**

Invite to Workshop Track

**Comment:**

Predicting graphs is an interesting and important direction, and there exist essentially no (effective) general-purpose techniques for this problem.  The idea of predicting nodes one by one, though not entirely surprising, is interesting and the approach makes sense. Unfortunately, I (and some of reviewers) less convinced by evaluation:

-  For example, evaluation on syntactic parsing of natural language is very weak. First of all, the used metric -- perplexity and exact match are non-standard and problematic (e.g., optimizing exact match would largely correspond to ignoring longer sentences where predicting the entire tree is unrealistic).  Also the exact match scores are very low (~30% whereas 45+ were achieve by models back in 2010).

- A reviewer had, I believe, valid concerns about comparison with GrammarVAE, which were not fully addressed.

Overall, I believe that it is interesting work, which regretfully cannot be published as a conference paper in its current form.

+ important / under-explored problem
+ a reasonable (though maybe not entirely surprising / original) approach
- issues with evaluation